# TRAIN NEURAL NETWORK BY EMBEDDING SPACE PROBABILISTIC CONSTRAINT

**Kaiyuan Chen**
Department of Computer Science
University of California
Los Angeles, CA 90095, USA
chenkaiyuan@ucla.edu

**Zhanyuan Yin**
Department of Mathematics
University of California
Los Angeles, CA 90095, USA
yinzhanyuan1999@ucla.edu

## ABSTRACT

Using higher order knowledge to reduce training data has become a popular research topic. However, the ability for available methods to draw effective decision boundaries is still limited: when training set is small, neural networks will be biased to certain labels. Based on this observation, we consider constraining output probability distribution as higher order domain knowledge. We design a novel algorithm that jointly optimizes output probability distribution on a clustered embedding space to make neural networks draw effective decision boundaries. While directly applying probability constraint is not effective, users need to provide additional very weak supervisions: mark some batches that have output distribution greatly differ from target probability distribution. We use experiments to empirically prove that our model can converge to an accuracy higher than other state-of-art semi-supervised learning models with less high quality labeled training examples.

## 1 INTRODUCTION

Probability is an abstract measure on how a certain event occurs independent of features of the events. Knowing how likely a certain event occurs, people leverages such prior knowledge to their decision making. For example, doctors know certain diseases are rare, even if they are told in terms of probabilities instead of "training examples". Based on this knowledge, they make less predictions on these diseases than those common ones. *Do neural networks behave in a similar way?* Unfortunately, the answer is no. When we train a multi-layer perceptron(MLP) for MNIST classifier (LeCun et al. (1998)) with limited labelled examples, the output distribution can be extremely biased in favor of some of the labels. In Figure 1a, we compare the predicted number of labels with ground truth. While the training accuracy is 1.0, the model clearly overfits to those training examples and leave labels between training data points undefined in high dimensional feature space. As we plot the last hidden layer of a MLP trained with 50 labelled MNIST data as shown in Figure 1b, we find neural networks fail to learn the decision boundary correctly from a limited number of examples.

Thus, it is natural to consider introducing output label probability distribution as higher order knowledge when we train neural networks. Different from traditional logical constraints (Xu et al. (2018)) or functional constraints (Stewart & Ermon (2016), we propose a novel embedding space probabilistic constraint. Because of the sparsity of high dimensional feature space with only a few labeled examples, we perform our probabilistic constraint on neural network's embedding space, which is constructed unsupervisedly by projecting data into low dimensional space through autoencoder. Based on observation by Xie et al. (2016), Zhang et al. (2016), embedding space preserves information of separations of different label clusters. In the embedding space, we pool softmax activation

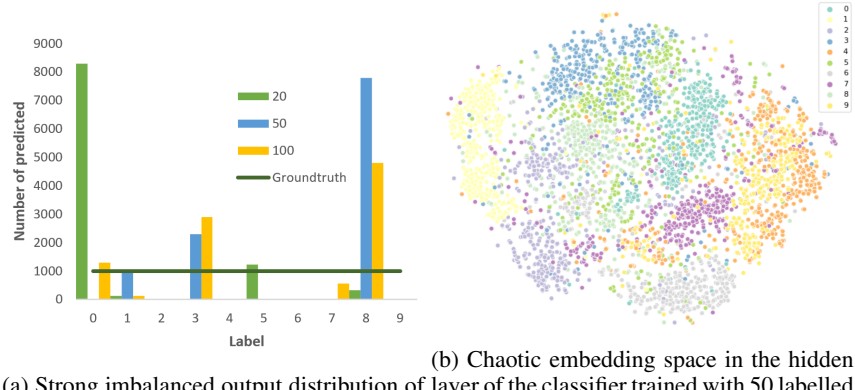

(b) Chaotic embedding space in the hidden
(a) Strong imbalanced output distribution of layer of the classifier trained with 50 labelled
labels when training set is limited examples

Figure 1: Limited training data cannot train neural networks to learn accurate decision boundaries

outputs and optimize towards target distribution. By training with very few high quality labelled examples and marking on batches that have output distribution greatly different from target probability distribution, we use experiments to empirically prove that our model can converge to a high accuracy faster than state-of-art semi-supervised learning methods.

## 2 RELATED WORKS

**Weak Supervision**   Current supervised representation learning algorithms have gained great success on various tasks in computer vision (He et al. (2017), Kostrikov et al. (2018)), natural language processing (Trivedi et al. (2018), Athiwaratkun et al. (2018)) with little domain knowledge, but they require large quantity and high-quality labels for training. Thus, there is a growing trend of research that address this problem by transferring knowledge learned from different datasets (Azizzadenesheli et al. (2018), Shen et al. (2017)) or introducing higher level knowledge.

In this work, we consider incomplete weak supervision problem (Zhou (2017)). A typical incomplete supervision problem (Chapelle et al. (2006)) is formulated as following: with a dataset $\{\mathbf{X}, Y\}$ that consists of labeled dataset $X_1 = \{\mathbf{X_1}, y_1\}$ and unlabeled dataset $X_2 = \{\mathbf{X_2}, y_2\}$, where $\{y_2\}$ is not visible during training. $|X_1| \ll |X_2|$. This problem can usually be tackled by state-of-art semi-supervised learning algorithms like AtlasRBF (Pitelis et al. (2014)), Neural Rendering Model (Ho et al. (2018))or LadderNet (Rasmus et al. (2015) or using novel approaches such as logical constraints (Xu et al. (2018)). While they still rely on certain amount of high quality labeled data, while in this work, we further decrease the number of labeled data needed for convergence.

**Learning With Constraints**   Learning with constraints takes various higher order domain knowledge into the optimization of neural networks. Based on domain knowledge, different constraints are effective on different tasks. For example, Pathak et al. (2015) uses linear constraints on the output space and optimizes the training objective as a biconvex optimization for linear models to perform dense pixelwise semantic segmentation. Frameworks such as semantic loss by Xu et al. (2018) and logical loss by Hu et al. (2016) specify logic rules when training neural networks. Stewart & Ermon (2016) propose a novel framework that one can learn physical or causal relationship without labels. In this work, we consider the case where limited labeled examples lead to biased output distribution. Different from these arithmetic or logical constraints, we consider placing an output probability constraint.

## 3 EMBEDDING SPACE PROBABILISTIC CONSTRAINT

In this section, we state our problem formulation and describe the proposed algorithm and architecture for this problem.

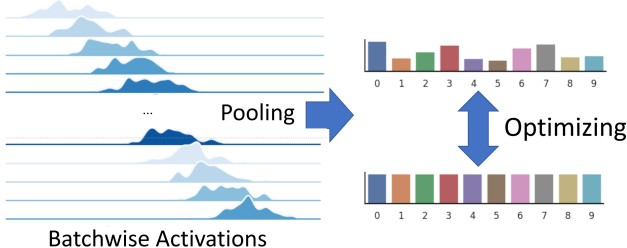

Figure 2: Flowchart of our probability constraint

**Higher-order Knowledge Formulation** Based on the incomplete weak supervision defined in Section 2, we specify our introduced higher order knowledge. We assume from our domain knowledge of output probability distribution, the model can acquire a set $Q = \{(k, \mathbb{P}(Y = k) + \epsilon\}_{k \in \{Y\}})$. One thing to note is that domain knowledge distribution $Q$ does not necessarily cover all $k \in Y$. We use $\epsilon \in \mathbb{R}$ drawn from Gaussian distribution such that $\epsilon \sim \mathcal{N}(0, \sigma^2)$ to reflect the variance of human domain knowledge from true $Y$. We use $\sigma = 0.05$ throughout this paper. We need a training algorithm $A(\{\mathbf{X}_1, y_1\}, \mathbf{X_2}, Q\})$ such that it trains a multi-layer perceptron $f(x) = \sigma(W_m...\Phi(W_2\Phi(W_1 x)))$, where $W_i$ is model's weights, $\Phi$ is an nonlinear function like ReLU, and $\sigma$ is the softmax function. This algorithm minimizes the loss function $\ell : \{Y, f(\mathbf{X})\} \to \mathbb{R}$.

**Batchwise Probability Constraint** Following from this problem formulation, we define a loss term $\ell_c : \mathbb{R}^2 \to \mathbb{R}$ to regulate the output distribution. We regard a single update batch $\{\mathbf{X}', Y'\} \in \{X, Y\}$ with size $c$ as the unit of output probability distribution. Inspired by Liang et al. (2017) and Hendrycks & Gimpel (2016), the activation of final softmax layer of classifier $\sigma(\cdot)$ can reflect neural network's confidence towards a certain label. Instead of performing counting the arguments of the maxima for all labels, which is not inefficient, we consider calculating the mean pooling of all the activation outputs(Wang et al. (2018)). It can be written mathematically as $\frac{1}{c} \sum_{i=0}^{c} f(\mathbf{X}_i')$. Here we use $f_k$ to denote the softmax activation of the $k$th label. This potentially improves the accuracy of detecting low confidence or out-of-distribution examples. A basic flowchart of our mechanism can be found in Figure 2.

It is natural to use Kullback-Leibler (KL) divergence as a metric of our output distribution different from the reference domain knowledge probability distribution $Q$, that is,

$$\ell_c = -\frac{1}{c} \sum_{(k,q) \in Q} \sum_{i=0}^{c} f_k(\mathbf{X}_i') \log \frac{qc}{\sum_{i=0}^{c} f_k(\mathbf{X}_i')} \tag{1}$$

One may notice the probability of labels in the batch does not always reflect the domain knowledge distribution $Q$. That is, $|\mathbb{P}(Y' = k) - Q(k)| > \epsilon$ for some $\epsilon > 0$. For the simplicity of this work, we assume additional but very weak supervision on identifying some of those batches and using different but noisy batch probability distribution. However, this supervision can be easily done through at-a-glance (abriel Ryan (2018)) supervision or auto-regressive algorithms similar to Reed et al. (2017). Our proposed algorithm and its convergence analysis can be found in Appendix A.

**Constraint on Embedding Space** In order to use existing unlabeled data to draw decision boundaries, we propose to jointly optimize this probability constrained classifier with an embedding space regularizer. Embedding is a lower dimensional form that structurally preserves data from original hyper-dimensional space. In our case, we treat a hyperparameter $i$th hidden layer of perceptron $E(x)$ as our embedding space, where $E(x) = W_i\Phi(...\Phi(W_2\Phi(W_1 x)))$ and $f(x) = \sigma(W_m(\Phi(W_{m-1}(...\Phi(W_{i+1}))) \circ E(x)$, where the dimension of $E(x)$ should be much smaller than dimension of input $x$. Zhang et al. (2016) propose using unsupervised loss can preserve information of separations between different label clusters. Thus, we adopt the structure of decoder of autoencoder and define a multi-layer neural network $D(\cdot)$ as a decoder of our embedding space. For a single batch $\{\mathbf{X}', Y'\}$, our loss function for training a separation-preserving embedding

space by reconstructing the original input, that is,

$$\ell_r = \frac{1}{c} \sum_{i=0}^{c} ||\mathbf{X}' - D(E(\mathbf{X}'))||_2$$

**General Framework**   Our proposed method uses unsupervised loss $\ell_r$ to construct an embedding in low dimensional space, uses limited labeled data to identify the cluster location in the embedding space by original classification loss $\ell_{original}$, and uses domain knowledge of output probability distribution to determine the actual decision boundaries. Then our updating loss function is

$$\ell = \lambda_1 \ell_{original} + \lambda_2 \ell_r + \lambda_3 \ell_c$$

, where $\lambda_1$, $\lambda_2$ and $\lambda_3$ are hyperparameter constants.

# 4   EVALUATION

**Experiment setup**   We evaluate our proposed embedding space probabilistic constraint in semi-supervised learning setting. Using the similar base multilayer perceptron model as in Rasmus et al. (2015) and Xu et al. (2018) All the experiments are repeated five times with different seeds. We add an additional embedding layer with width 40, and the decoder has a symmetric architecture as the feed forward neural network.

**Model Description**   To guarantee that our comparison focuses on output probability distribution instead of one single instance's label, we train our models with batch size 128. We experiment our model under different level of constraints. Datasetwise probability constraints assumes the target output should be all $10\%$, and the noisy datasetwise probability constraints adds a random noise drawn from $\mathcal{N}(0, 0.3)$ to simulate user's knowledge. Also, we use batchwise probability constraint, which assumes we know the probability of labels in every batch, as an upper bound for our algorithm. We compare our model with other state-of-art semi-supervised learning models (Pitelis et al. (2014), Rasmus et al. (2015)), and logical constraint model (Xu et al. (2018). Since we require more human supervision than other semi-supervised learning models, we use their results to demonstrate our model can converge to high accuracy with much less high quality labeled examples. We also choose other baselines models without both losses and without embedding loss to show the benefit of our architecture.

| Accuracy/# of labelled per class | 3 | 5 | 10 | all |
|---|---|---|---|---|
| AtlasRBF (Pitelis et al. (2014)) | $73.58 \pm 0.95$ | $84.28 \pm 0.21$ | $91.54 \pm 0.13$ | $98.20 \pm 0.25$ |
| Ladder Net (Rasmus et al. (2015)) | $79.39 \pm 0.60$ | $93.67 \pm 1.42$ | $97.69 \pm 0.25$ | $99.01 \pm 0.22$ |
| Baseline: MLP | $53.52 \pm 0.07$ | $64.07 \pm 0.19$ | $73.24 \pm 0.13$ | $98.82 \pm 0.03$ |
| Semantic Loss (Xu et al. (2018)) | $75.36 \pm 1.02$ | $82.53 \pm 1.39$ | $96.03 \pm 1.39$ | $98.53 \pm 1.39$ |
| Baseline: MLP with constraint | $78.39 \pm 3.76$ | $92.97 \pm 3.28$ | $96.82 \pm 2.39$ | $97.01 \pm 3.03$ |
| Datasetwise Probability Constraint(noisy) | $82.12 \pm 1.34$ | $96.05 \pm 0.05$ | $96.32 \pm 0.24$ | $97.85 \pm 0.93$ |
| Datasetwise Probability Constraint | $84.93 \pm 1.08$ | $97.65 \pm 0.05$ | $97.32 \pm 0.24$ | $98.45 \pm 0.84$ |
| Batchwise Probability Constraint | $95.87 \pm 0.48$ | $97.67 \pm 0.39$ | $98.67 \pm 0.74$ | $98.99 \pm 0.21$ |

Table 1: Semi-supervised learning on MNIST dataset

| Accuracy/# of labelled per class | 3 | 5 | 10 | all |
|---|---|---|---|---|
| Baseline: MLP | $49.72 \pm 0.12$ | $55.72 \pm 0.11$ | $68.64 \pm 0.10$ | $86.32 \pm 0.21$ |
| LadderNet (Rasmus et al. (2015)) | $69.87 \pm 0.43$ | $74.54 \pm 0.32$ | $81.87 \pm 0.15$ | $90.57 \pm 0.28$ |
| Semantic Loss (Xu et al. (2018)) | $70.24 \pm 1.30$ | $75.41 \pm 0.21$ | $83.25 \pm 0.23$ | $89.92 \pm 0.32$ |
| Batchwise Probability Constraint | $81.15 \pm 0.02$ | $85.15 \pm 0.85$ | $88.15 \pm 0.92$ | $89.01 \pm 1.20$ |

Table 2: Semi-supervised learning on FASHION dataset

# 5   DISCUSSION

In this paper, we present an algorithm that constraints output probability distribution in an embedding space. Our result shows with a little more supervision than normal semi-supervised learning

| Accuracy/# of labelled per class | 200 | 400 | all |
|---|---|---|---|
| Baseline: CNN | $59.72 \pm 0.12$ | $76.64 \pm 0.10$ | $90.32 \pm 0.21$ |
| Semantic Loss(Xu et al. (2018)) | $71.24 \pm 1.30$ | $79.41 \pm 0.21$ | $89.93 \pm 0.45$ |
| Batchwise Probability Constraint | $75.15 \pm 0.52$ | $86.15 \pm 1.03$ | $88.15 \pm 0.92$ |

Table 3: Semi-supervised learning on CIFAR 10 dataset

algorithms, we need far fewer high quality training examples to reach high accuracy. Thus, we conclude jointly optimizing the output constraint with hypothesis can draw a decision boundary with smaller labelled training data than other state-of-art methods.

Our focus is to show the power of a very weak labelling method without high quality labelling technique. We leave it as a future research direction to design an auto-regressive algorithm that requires less supervisions. In addition, since our formulation sums all the activation functions together as a measure of confidence instead of counting-based probabilities, this allows us to use it as a future direction on confidence of classification in semi-supervised learning setting.

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

# A  ALGORITHM

In Bootstrap phase, we require users to provide an entry supervision mark $\{\Theta\}$. It marks entries that has unusually high probabilities of occurrence from target probability distribution. We need this extra supervision because reliance on model's own probabilistic judgement might converge to a wrong model. Feeding additional but weak supervision can give model enough information to judge if current probability distribution has such unusual batches. We also claim that we can bound it by concentration inequalities, as shown in Theorem 1.

The probability of this mark happens can be bounded by Theorem 1.

**Theorem 1** *For a single batch $\{X, y\}$ with size c, on a label k with domain knowledge output probability $Q(k)$. For all $\epsilon > 0$, we have*

$$\mathbb{P}(\frac{1}{c}\sum_{i=0}^{c}(\mathbb{P}(|Y=k)-Q(k)| \geq \epsilon) \leq 2\exp(-2c\epsilon^2)$$

**Proof of Theorem 1.1** *This claim directly follows from Hoeffding inequality, which we state as following:*
*For independent bounded random variables $X_i \in [a_i, b_i]$ and for all $t > 0$, which has $a \leq X \leq b$, for all $\epsilon$, we have*

$$\mathbb{P}(\frac{1}{n}\sum_{i=1}^{n}X_i - \mathbb{E}[\frac{1}{n}\sum_{i=1}^{n}X_i] \geq \epsilon) \leq 2\exp(\frac{-2n^2t^2}{\sum_{i=1}^{n}(a_i-b_i)^2})$$

*The proof can be found in Shalev-Shwartz & Ben-David (2014).*

*In our case, our random variable is $\mathbb{P}(|Y=k)$, ranging from $[0, 1]$, while the target distribution, by higher knowledge, is $\mathbb{E}[\mathbb{P}(|Y=k)] = Q(k)$. With batch size c, we have*

$$\mathbb{P}(\frac{1}{c}\sum_{i=0}^{c}(\mathbb{P}(|Y=k)-Q(k)| \geq \epsilon) \leq 2\exp(-2c\epsilon^2)$$

We state our bootstrap algorithm in Algorithm 1. When we found a marked batch with higher probability, we redistribute our new target distribution based on its original target distribution. When an output probability is marked, we rescale the target distribution of the label $q_{marked}$ to adapt the higher-than-usual probability by Equation 2. A detailed derivation can be found in Appendix B.2.

$$q_{new} = q_{marked} + \sqrt{\frac{\log(1/\epsilon)}{2c}} \tag{2}$$

---

**Algorithm 1:** Bootstrap Probabilistic Constraint

---

**Data:** Training batch $\{\mathbf{X}, y\} \in \mathbb{R}^{c \times m} \times \mathbb{R}^n$, entry supervision mark $\{\Theta\}$, error threshold $\epsilon$,
   Domain knowledge output distribution $Q \in \mathbb{R}^n$, neural network $f$
Feed Forward $f(\mathbf{X})$ and pool output activations by $\frac{1}{c}\sum_{i=0}^{c}f(\mathbf{X}_i)$ ;
**for** $i \leftarrow 0$ **to** $m$ **do**
 **if** *this batch is marked* **then**
  $k = \sqrt{\frac{\log(1/\epsilon)}{2c}}$;
  $q_i = Q_i + k$;
 **else**
  $q \leftarrow Q_i - \frac{k}{m-|\Theta|}$;
 **end**
**end**
Train the network $f$ with $\{X, y, q\}$ by Equation 3;

---

The boundary between bootstrap phase and auto-regressive phase is a hyper-parameter. In this work, we use validation set to observe the accuracy of current model. When the accuracy on validation set is larger than 70%, we enter auto-regressive phase. A convergence plot example can be found in Figure 3.

---

**Algorithm 2:** Auto-Regressive Probabilistic Learning

---

**Data:** Training batch $\{\mathbf{X}, y\} \in \mathbb{R}^{c \times m} \times \mathbb{R}^n$, error threshold $\epsilon$, Domain knowledge output
     distribution $Q \in \mathbb{R}^n$, neural network $f$
Feed Forward $f(\mathbf{X})$ ;
$\Omega \leftarrow$ pool output activations by $\frac{1}{c} \sum_{i=0}^{c} f(\mathbf{X}_i)$ ;
$\Theta \leftarrow \emptyset$;
**for** $i \leftarrow 0$ **to** $m$ **do**
    **if** $|\Omega_i - Q_i| > \epsilon$ **then**
        | $\Theta \leftarrow \Theta \cup \Omega$;
    **else**
**end**
$q \leftarrow Q \wedge \neg\Theta$;
Train the network $f$ with $\{X, y, q\}$ by loss function from Equation 3;

---

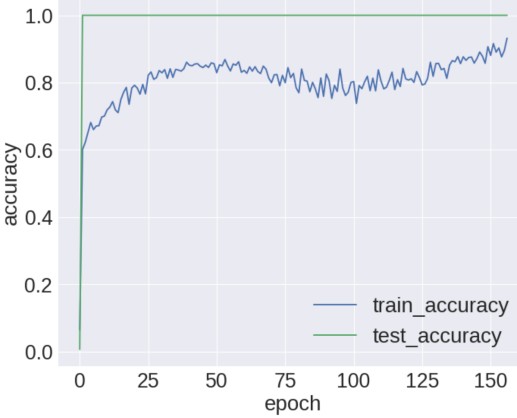

Figure 3: Single run of algorithm on convergence when trained with 30 labeled examples

## B  DERIVATIONS

### B.1  DERIVATIONS FOR EQUATION 1

**Lemma 1** *Let f be a feed forward neural network with softmax as the last layer's activation function.
Then given a batch $\{X, y\} \in \mathbb{R}^{c \times m} \times \mathbb{R}^n$, the mean-pooling $\Omega = \frac{1}{c} \sum_{i=0}^{c} f(\mathbf{X}_i)$ is a probability
distribution.*

**Proof of Lemma 1.1** *Without loss of generality, we write function $f(x)$ as $\sigma(\cdot) \circ g(x)$ for some
function $g(x)$. Then for all $x \in X$, $f(x)$ is in range of $\sigma(\cdot)$, that is, $\exists y \in R^n$, $f(x) = \sigma(y)$. Then
for a vector $z \in \mathbb{R}^n$, for any label $i$,*

$$\sigma(z)_i = \frac{e^{z_i}}{\sum_{k=1}^{n} e_i^z}$$

*Then we perform mean-pooling on $\sigma(y)_i$, we have*

$$\Omega_i = \frac{1}{c} \sum_{i=0}^{c} f(\mathbf{X}_i) = \frac{1}{c} \sum_{i=0}^{c} \sigma(y_i) = \frac{1}{c} \sum_{i=0}^{c} \frac{e^i}{\sum_{k=1}^{n} e_i^y}$$

*The pooled output activation $\Omega_i$ is trivially larger than 0, and since $\sigma(z)_i < 1$ for all $z$,
$mean(\sigma(z)_i) < 1$. Since $\sum_i \Omega_i = 1$, $\frac{1}{c} \sum_{i=0}^{c} f(\mathbf{X}_i) = \frac{1}{c} c = 1$. Then we conclude it fullfills
all the axiom of a probability distribution.*

Since $\Omega$ is a probability distribution by lemma 1, we can apply KL divergence to $\Omega$ from our target
distribution Q. With specified constraints set $K$, with $f_k(x)$ for the specific dimension of $k \in K$,

we have

$$
\begin{aligned}
\ell_c &= KL(\Omega||Q) \\
&= -\sum_{k \in K} \Omega(k)\left(\frac{Q(k)}{\Omega(k)}\right) \\
&= -\sum_{k \in K} \frac{1}{c}\sum_{i=0}^{c} f_k(\mathbf{X}_i)\left(\frac{Q(k)}{\frac{1}{c}\sum_{i=0}^{c} f_k(\mathbf{X}_i)}\right) \\
&= -\frac{1}{c}\sum_{k \in K}\sum_{i=0}^{c} f_k(\mathbf{X}'_i)\log\frac{Q(k)c}{\sum_{i=0}^{c} f_k(\mathbf{X}'_i)}
\end{aligned}
$$

## B.2 DERIVATIONS FOR EQUATION 2

For a single training example $x^j \in \mathbf{X}$, the features for $x^j = \{x_1^j, x_2^j...x_n^j\}$ are dependent, so we cannot apply statistical bounds by entries. However, when we compare training examples in the same batch, that is, $x_1^j$ and $x_1^{j+1}$, they are independent. As a result, we can still use the Proof of Theorem 1.1. When we would like to bound

$$
\mathbb{P}\left(\frac{1}{c}\sum_{i=0}^{c}(\mathbb{P}(|Y=k)-Q(k)| \geq d\right) \leq \epsilon
$$

for some error $d$, we rearrange

$$
\mathbb{P}\left(\frac{1}{c}\sum_{i=0}^{c}(\mathbb{P}(Y=k)-Q(k) \geq d\right) \leq \exp(cd^2) \leq \epsilon
$$

that is,

$$
d = \sqrt{\frac{\log(1/\epsilon)}{2c}} \tag{3}
$$

In this case, we find a safe margin $d$ that controls the tradeoff between human supervision and training batch accuracy. Directly applying the upper bound is empirically fine when we perform our experiments.

## C IMPLEMENTATION DETAILS

This section includes main implementation details not included in main text.

### C.1 DATASET DESCRIPTION

**MNIST**  MNIST dataset (LeCun et al. (1998)) is a dataset of handwritten digits, which has 60000 training images and 10000 testing images. For each of the image, it is a grey-scale $28 \times 28$ matrix that belongs to 10 classes from 1 to 10.

**FASHION**  a dataset of clothes that possess similar structure as MNIST, which has 60000 training images and 10000 testing images. For each of the image, it is a grey-scale $28 \times 28$ matrix that belongs to 10 classes for different types of clothes.

**CIFAR-10**  CIFAR-10 (Krizhevsky (2009)) is a dataset that contains colored images with size $32 \times 32$. Each image belongs to one of ten classes like dog, cat, car. The training set has 50000 images and the testing set has 10000 images.

### C.2 DATASET PREPARATION

**Selection**  In order to make the probability the same for all classes, we keep the same number of images among 10 classes. We choose the number as the minimum number of examples out of all

classes. For example, for MNIST dataset, it has numbers of examples 5923, 6742, 5958, 6131, 5842, 5421, 5918, 6265, 5851 respectively from 0 to 1. Since 5421 is the minimum for all labels, we choose 5421 as the number of examples we use. Based on this threshold, we choose the training examples randomly from the dataset.

To make examples from different classes balanced, we choose randomly (number of labeled examples/number of classes) number of examples.

**Processing**    We first normalize all images scale from 0 to 1. Then we have $50\%$ of chance to perform two operations on training examples to prevent overfitting of the training set: adding Gaussian noise and cropping. For Gaussian noise, we add to the image another matrix of random noise drawn from Gaussian distribution with mean 0 and standard deviation 0.3. We crop the image by three pixels. For example, if the original image has pixel $28 \times 28$, our cropped image has $25 \times 25$.

## C.3    ARCHITECTURE

**Multi-layer Perceptron**    We evaluate our proposed embedding space probabilistic constraint in semi-supervised learning setting. Using the similar base multilayer perceptron model as in Rasmus et al. (2015) and Xu et al. (2018) with layer of the size 784-1000-500-250-250-250-40-10, except adding a layer with width 40 as embedding layer, we feed the output of the embedding layer to another multilayer deocoder with size 40-250-250-1000-784. We also perform batch normalization and dropout with a dropout rate $50\%$ to increase robustivity of our model.

**Convolution Neural Network**    To compare fairly with other state-of-art models, we adopt a basic 10-layer architecture similar to Xu et al. (2018) for classification. For every three layers, we use one convolution layer with activation function ReLU, one $2 \times 2$ max-pool layer with stride 2. We insert one layer of dropout with a dropout rate 0.5 and one batch normalization layer as we described in multi-layer perceptron. For the decoder, we use a symmetric decoder. For every three convolution layers, we use a upsampling layer. We insert batch normalization layer and dropout layer between them. The embedding space has a vector with dimension 50.

## C.4    TRAINING

Detailed hyperparameter is as following: we use Stochastic Gradient Descent(SGD) to update our neural network with a learning rate $10e-4$. For simplicity, we choose $\lambda_1$, $\lambda_2$ and $\lambda_3$ to be 1. A rule of thumb is to make $\lambda_1 \ell_{original} \approx \lambda_2 \ell_c$. When they possess similar values, they can converge to the solution quickly. The convergence of our algorithm can be shown in Figure 3.

In the cold start session, with batch size 128, we choose to find batch output probabilities $p \notin [0.02, 0.25]$. Empirically this covers $98\%$ of data, we can safely set them with probability 0.1, while the rest but obvious data, we set them a manual probability, in our case, we set them 0.02 and 0.28.

