# OpenReview forum: "Train Neural Network by Embedding Space Probabilistic Constraint "
_ICLR.cc/2019/Workshop/LLD — LLD 2019_

### Official Review · AnonReviewer2 · 2019-04-08
**The work is highly relevant to the workshop. Results are good. Writing is sufficiently clear. Maths and notation are not.**

**Rating:** 3
**Confidence:** 2

**Review:**

1/ Introduction
The first sentence is embarrassing as 'probability' has a mathematical definition.
The problem target precisely the workshop topic.

2/ Related work
Ok.

3/ Embedding...
There are inconsistencies in the notations and many typos.
Is Q a 2d vector ? Why is it compared to P(k) later ?

4/ Evaluation
Results are competitive.

---

### Official Review · AnonReviewer3 · 2019-04-10
**Nice performance, but needs some work before publication**

**Rating:** 2
**Confidence:** 2

**Review:**

This paper introduces a method to perform semi-supervised learning with deep neural networks.  The model is trained on a few labeled examples and the expected label distribution to achieve relatively high accuracy, given the small training size.  The problem is motivated well and provides a clear introduction and background of related work. Included figures provide useful reference for the paper and the experiments demonstrate that the approach works well.

I found that the assumption that the probability distribution of true labels could be well known to not be very realistic.  I think it would be quite interesting to see how your approach performs when faced with different levels of error (including non-gaussian error) in the probability distribution. The assumption that out-of-distribution batches are identified through weak supervision seems like a large caveat to me.

When discussing the Model Description in Section 4, you say: “To guarantee that our comparison focuses on output probability distribution instead of one single instance’s label, we train our models with batch size 128.” This statement seems overly specific to me. How does a batch size of 128 specifically guarantee that your comparison focuses on the output probability distribution? Would any sufficiently large batch size work? How do you find sufficiently large batches?

Overall, the paper presents an architecturally simple solution that seems to work well to improve accuracy in a semi-supervised setting.

The paper has quite a few grammatical and formatting errors. It should be thoroughly read and edited before publication (I noted some of the grammatical issues below that I hope will be helpful).

Editing Suggestions:
- There appear to be several instances where the spacing between words and citations is incorrect. There are also a few times where parenthesis around citations were not properly closed.
- missing closing parenthesis in “constraints (Stewart & Ermon (2016),”
- check spacing in citations - Ho et al.
- It would be nice to see a reference to prior work which introduces “higher level knowledge” in the last sentence of the “Weak Supervision” paragraph.

Small notes on language that may make it more readable:
- “neural networks fails to learn the decision boundary correctly from limited number of examples” -> “neural networks fail to learn the decision boundary correctly from a limited number of examples”
- “we perform our probabilistic constraint on neural network’s embedding space,” -> “we perform our probabilistic constraint on the neural network’s embedding space,”
- “low dimensional space through autoencoder.” -> “low dimensional space through (an or the) autoencoder.”
- “our model can converge to an high accuracy faster than” -> “our model can converge to a high accuracy faster than”
- “require large quantity and high-quality labels for training”-> “require a large quantity of high-quality labels for training”
- “can reflect neural network’s confidence towards certain label.” -> “can reflect the neural network’s confidence towards a certain label.”
- “Instead of performing counting the arguments of the maxima for all labels, which is not inefficient,” -> “Instead of counting the arguments of the maxima for all labels, which is not inefficient,”
- “We add additional embedding layer with width 40,” -> “We add an additional embedding layer with width 40,”
- “We experiment our model under different level of constraints. “ -> “We experiment under different level of constraints. “
- “that constraints output probability distribution in an embedding space.” -> “that constrains the output probability distribution in an embedding space.”
- “we need far less high quality training examples to reach high accuracy” -> “we need far fewer high quality training examples to reach high accuracy”
- “Thus, we conclude jointly optimizing” -> “Thus, we conclude that jointly optimizing”
- “It can me written mathematically” -> “It can be written mathematically”
- “Thus, we adopt the structure of decoder of autoencoder and “ -> “Thus, we adopt the structure of the decoder of an autoencoder and“
- “Our proposed method uses unsupervised loss” -> “Our proposed method uses an unsupervised loss”
- “and uses domain knowledge of output probability distribution to determine the actual decision boundaries.” -> “and uses domain knowledge of the output probability distribution to determine the actual decision boundaries.”

---

### Official Review · AnonReviewer1 · 2019-04-10
**Leveraging additional supervision to handle label distribution shift**

**Rating:** 3
**Confidence:** 2

**Review:**


This paper tries to incorporate label distribution into model learning, when a limited number of training instances is available.
Intuitively, the output label distribution could be wrongly biased, and the prior information like label distribution could be helpful.
To handle this problem, the authors propose two different techniques, the first regulate the output distribution and the second regularize the constructed representation.
Performance comparison demonstrated the effectiveness of the proposed method when only a limited number of instances are available.

I think the studied problem is interesting and the proposed solution is novel and reasonable.
My main concern is about the assumption of the algorithm.
The proposed learning algorithm assumes that the algorithm can access a relative accurate label distribution, and the output distribution regularization depend on this term.
But for real world applications, it could be hard to get such knowledge, since in order to get the required annotation (as described in Appendix), the user needs to have a good understanding of the real distribution or annotate all instances in that batch.
Besides, I noticed that in the experiment results, the proposed method sometimes achieves worse performance than baselines when all training data is available.
This phenomenon seems to me implies that the proposed method cannot fully leverage the additional information, as intuitively, with more information, it should perform better.

---

### Decision · Program_Chairs · 2019-04-16
**Acceptance Decision**

Accept